# Factors Influencing the Choice to Advise for or against COVID-19 Vaccination in the Physicians and Dentists of an Italian Region

**DOI:** 10.3390/vaccines10111793

**Published:** 2022-10-25

**Authors:** Eleonora Marziali, Alberto Lontano, Luca Regazzi, Andrea Paladini, Leonardo Villani, Giovanna Elisa Calabrò, Gianfranco Damiani, Patrizia Laurenti, Walter Ricciardi, Chiara Cadeddu

**Affiliations:** 1Section of Hygiene, University Department of Life Sciences and Public Health—Università Cattolica del Sacro Cuore, 00168 Rome, Italy; 2Department of Women, Children and Public Health—Fondazione Policlinico Universitario “A. Gemelli” IRCCS, 00168 Rome, Italy

**Keywords:** vaccine hesitancy, information sources, physicians, dentists, COVID-19 vaccines, Italy

## Abstract

Healthcare workers (HCWs), particularly physicians, are a relevant and trusted source of information for patients, especially when health-related choices such as vaccination are concerned. Between July and November 2022, we administered a web-based survey to physicians and dentists living in the Latio region of Italy to explore whether their background might affect their willingness to recommend the COVID-19 vaccination to their patients (RCVtoPat) and their relatives (RCVtoRel). We performed a multivariable logistic regression to study the association between the two outcomes (RCVtoPat and RCVtoRel) and their potential determinants in our sample (*n* = 1464). We found that being a dentist, an increasing fear of COVID-19, and having been previously vaccinated against flu are positively associated with both RCVtoPat and RCVtoRel, while a better self-rated knowledge of COVID-19 vaccines is associated only with RCVtoRel. No role was found for age, sex, civil status, education level, information sources, previous SARS-CoV-2 infection, and chronic diseases. A sub-group analysis of physicians alone (*n* = 1305) demonstrated a positive association with RCVtoRel of being specialized in diagnostic/therapeutic services and a negative effect on RCVtoPat of being trained in general practice. We provide useful insights about the factors that should be addressed to ensure HCWs exert a positive influence on their patients and communities.

## 1. Introduction

As of 23 September 2022, more than 22 million laboratory-confirmed cases of SARS-CoV-2 infection have been recorded in Italy, with approximately 415,000 healthcare workers (HCWs) being affected, and more than 170,000 deaths from COVID-19 [1]. To deal with the COVID-19 pandemic, a mass vaccination campaign was launched in Europe [2]. Specifically, the Italian vaccination campaign started on 27 December 2020, with the aim of covering 1.5 million HCWs by the first trimester of 2021 [3]. As the vaccine supply was not immediately available to immunize the whole population, the national vaccination plan prioritized HCWs, nursing home residents and workers, and people aged ≥ 80 [4,5]. Since the approval of the first COVID-19 vaccine in the European Union/European Economic Area (EU/EEA) in December 2020, the body of evidence regarding vaccine effectiveness and the population impact has been increasing. Data from the real-world usage of COVID-19 vaccines have confirmed the findings of clinical trials [6] and also showed a high vaccine effectiveness against PCR-confirmed SARS-CoV-2 infection, especially for the first circulating variants, significantly reducing the viral load [7]. Nevertheless, for many individuals these demonstrated benefits of vaccines are not sufficient to embrace vaccination whole-heartedly, resulting in low adherence to vaccination [8].

Vaccine hesitancy (VH) is defined as the refusal, delay, or acceptance of vaccines with doubts about their usefulness and safety [9], and the World Health Organisation (WHO) declared vaccine hesitancy as one of the ten biggest threats to global health in 2019 [10]. HCWs are a primary source of information about vaccinations [11,12] and play a crucial role in their patients’ vaccination compliance [13,14]. Besides their position at the forefront of the battle against COVID-19, physicians must be able to provide information to their patients and respond to their anxieties and concerns [15,16]. Nonetheless, a growing number of publications from various countries show that VH also exists among HCWs [17]; actually, recent reports suggest that many HCWs are also hesitant about or are delaying getting their COVID-19 vaccine [18,19,20,21]. VH among Italian HCWs is a topic already investigated; indeed, insufficient vaccination coverage in Italian health personnel is reported, considering other vaccine-preventable diseases recommended for the category [22,23,24,25]; for example, in a recent multicenter study in Italy, vaccination coverage among health-care workers was 77.3% for HBV [26], while among the 4483 HCWs enrolled in two large Italian hospitals, the influenza vaccination coverage was 32.5% [25,27,28,29,30,31]. In addition, HCWs’ hesitancy reflects their education and advanced practice scope; indeed, physicians and dentists are more likely to support vaccination than nurses and physiotherapists [27,28,29,30,31]. Similarly, a recent cross-sectional study on COVID-19 vaccine hesitancy and determinants in healthcare students reveals that nursing students and physiotherapy students are more hesitant than medical students [32].

The rationale of this study consists of analyzing the knowledge, attitudes, and behaviors regarding the COVID-19 vaccines of physicians and dentists from Lazio in order to investigate the main elements of the VH phenomenon in this group.

## 2. Materials and Methods

### 2.1. Study Design and Sample

This study is part of a more extensive cross-sectional study based on an online survey conducted between 22 July and 20 November 2021, in Italy among a sample of HCWs [33]. All adult physicians and dentists residing and working in the Lazio region who completed the survey were included in the present study and were analyzed in a post hoc analysis. Only Lazio, the region where Rome is located, was considered because our analysis involved also the Rome Provincial Order of Physicians and Dentists “OMCEO Rome”. In Italy, both physicians and dentists belong to the same professional association, as until 1980, the year in which the specific degree program in dentistry was established, dentists were physicians who decided to specialize in dentistry. We decided to specifically analyze the data on these professions because they represent a main point of reference for patients to get clinical advice and health information [34,35,36,37].

The questionnaire was developed through a literature review and was validated by a pilot involving 30 HCWs [30,38,39,40,41,42,43,44]. The online survey was created using the “Survey Monkey^®^” platform, developed by Momentive Inc., and the link was sent via newsletter to physicians and dentists by the professional association of the province of Rome (the Rome Provincial Order of Physicians and Dentists). The “OMCEO Rome” was the main form of dissemination of the questionnaire, however, the questionnaire also reached members of professional orders in other provinces of the Lazio region through word of mouth and through flyers spread during the national congress of the Italian Society of Hygiene, Preventive Medicine and Public Health. In order to fill out the questionnaire, all participants had to view the study description, approve the personal data management, and sign an informed consent form regarding the anonymous, voluntary, and unpaid nature of the study. The data collection was anonymous, and it is not possible to link the recorded information to the respondents.

The study was approved by the Ethics Committee of the “Fondazione Policlinico Universitario A. Gemelli—IRCCS” in Rome (Prot. No. 0021609/21 ID 4104) and was conducted according to Good Clinical Practice (GCP) standards and the requirements of the Declaration of Helsinki. The data collected were processed according to EU Regulation No. 2016/679 (GDPR), Legislative Decree No. 196/2003 “Code on the Protection of Personal Data” and the subsequent amendments, and all the current legislation on data processing and protection.

### 2.2. Questionnaire

The questionnaire consisted of 39 closed-ended questions divided into 5 sections (see Appendix) [45]: (1) sociodemographic characteristics and work information (age, sex, region of residence, marital status, education level, occupation and discipline, workplace, and impact of the COVID-19 pandemic on work activity); (2) health conditions (chronic diseases that result in insusceptibility to severe forms of COVID-19 and previous SARS-CoV-2 infection); (3) opinions and attitudes about vaccines, in general, and COVID-19 vaccines in particular (assessed through a 5-option Likert scale to rate the degree of agreement or disagreement); (4) intentions, behaviors, and attitudes regarding vaccination and the factors influencing these behaviors: information regarding previous influenza vaccination, propensity to vaccinate against COVID-19, and willingness to recommend vaccines to patients and family; and (5) knowledge on COVID-19 vaccine and sources of information used, perceived level of preparedness, topics they would like to explore, and desired channels of communication.

### 2.3. Statistical Analysis

A sub-sample of 1526 physicians and dentists working in the Lazio region was selected from the original sample of 2132 Italian healthcare workers who completed the survey [33].

A post hoc analysis was performed to explore the behavior of physicians and dentists towards advising the COVID-19 vaccination to their patients and family, as well as the possible determinants of such behavior. Two questions from the questionnaire were selected as the outcome variables (Q33. are you recommending the COVID-19 vaccine to your patients?, and Q34. are you recommending the COVID-19 vaccine to your family members?) and dichotomized to increase the interpretability, as shown in Table 1.

The other questions from Section 1, Section 2, Section 4 and Section 5 have been considered as the independent variables. The questions from Section 3 have been excluded from analysis, as they were extensively analyzed in a previous study [33].

Descriptive statistics (the relative frequency for the categorical variables, and mean and standard deviation for the continuous variables) were used to explore the distribution of the independent variables overall (Table 2) and with respect to Q33 (Table 3) and Q34 (Table 4), individually.

A multivariable logistic regression analysis was employed to evaluate and quantify the association between the independent variables and the two outcome variables. The logistic regression was applied by fitting a logit model for a binary response by the maximum likelihood. The independent variables to be included in the multivariable model have been selected on the basis of preliminary significance (*p* < 0.10) in logistic univariable models. Age has been included in the model despite the non-significance to account for potential confounding. The regression analysis was also repeated for a narrower sub-sample of only dentists working in the Lazio region and on a narrower sub-sample of only physicians working in the Lazio region, the latter including in the model two further independent variables (medical specialization and working setting). The respondents who preferred not to specify their sex (*n* = 5, 0.3%) were excluded from the logistic regression analysis, as they constituted an excessively small sub-group. The results of the logistic regression have been reported in the form of an odds ratio with 95% confidence intervals (Table 5 and Table 6). *p*-values equal to or below 0.05 were considered to be statistically significant.

All the statistical analyses were carried out using Stata software, version 15 (StataCorp LP, College Station, TX, USA).

## 3. Results

One thousand five hundred and twenty-six physicians and dentists consented to take part in the survey, of which 810 were men (53.1%) and 711 were women (46.6%), aged between 24 and 88. Their demographic information is reported in Table 2.

Among the HCWs, 76% declared to have received a flu jab in the previous vaccination campaign, while almost 99% had received the COVID-19 vaccine. Among the latter, 80% received the vaccine because they believed in vaccines as a public health tool to tackle epidemics/pandemics, almost 60% did so because of a sense of responsibility, 56% received it because they were in a high-risk profession, 45% wanted to protect their family, and 16% were at risk of developing a serious COVID-19 disease. Considering those who refused to get the COVID-19 vaccine (1% of all HCWs), almost 67% wanted further evidence of vaccine efficacy and safety, 56% were afraid of the possible side effects of the vaccine, almost 40% thought they were not at risk of developing severe COVID-19, 22% suffered from a medical condition that contraindicates the administration of the vaccine, while 5% considered COVID-19 to be a non-serious illness. Considering a score from 1 to 10, the impact of the fear of being infected by COVID-19 on the desire to be vaccinated had an average value of 7.0. Concerning the main sources which HCWs used to expand their knowledge of COVID-19 vaccines, 84.3% of them relied on institutional websites (such as the WHO or Italian Ministry of Health), 71.8% on scientific literature, 48% on webinars or updating courses, 33.4% on colleagues, 15.1% on newspapers or magazines, 9.6% on television or radio, and 9.4% on websites other than institutional ones. Considering a score from 1 to 10, the perception of one’s knowledge about COVID-19 vaccines had an average value of 7.5. Moreover, 68.7% of the sample declared they wanted further information on the safety and side effects of COVID-19 vaccines, 46.9% on the effectiveness of COVID-19 vaccines, 34.9% on the existing types of COVID-19 vaccines, while 12.9% affirmed they did not need any further information about COVID-19 vaccines. Seventy-three per cent and two tenths of the HCWs expressed the wish to gain further knowledge on COVID-19 vaccines by the means of an institutional website, 49.9% through updating courses, 40.3% through webinars, and 6.9% through podcasts. Regarding the tendency to recommend the COVID-19 vaccination to their patients, 91% of HCWs reported that they recommended the COVID-19 vaccine and 94% of them advised their relatives to take the vaccine. Moreover, no particular differences could be found within the variables considered, apart from for the health profession, work department among physicians, using institutional sources for information, and having received a flu vaccination in the previous vaccination campaign (Table 3). In particular, 12.4% of physicians did not recommend the vaccine to their patients compared to 4.1% of dentists that did not; interestingly, only 91.2% of general practitioners (GPs) recommended the vaccine to their patients, whereas the values are comparable for other medical specialties, spanning from 96.1% (clinical) to 99.4% (diagnostical and therapeutical). Those who used institutional sources tended to advice vaccination to patients more frequently (95.2%) then those who did not (87.2%). Moreover, 97.4% of those who received a flu vaccination in the previous vaccination campaign tended to recommend the COVID-19 vaccine to patients, compared to 83.3% of those who did not receive it. Among those advising the COVID-19 vaccine to their patients, the mean value of the score referring to their fear of being infected by COVID-19 was 7.2 out of 10, while those who did not reported a much lower average score (3.8 out of 10). Interestingly, those who recommended vaccination to patients and those who did not scored similarly when considering their knowledge about COVID-19 vaccines (7.5 vs. 7.1 out of 10).

Regarding the tendency to recommend the COVID-19 vaccination to their relatives, no particular differences can be found within the variables considered, but for one’s health profession, the work department among physicians, using institutional sources for information, having received a flu vaccination in the previous vaccination campaign, and receiving a previous diagnosis of COVID-19 (Table 4). Again, 11.3% of physicians did not recommend the vaccine to their relatives, compared to 4.3% of dentists, and only 89.8% of GPs recommended the vaccine to their relatives, whereas the values are comparable for other medical specialties, spanning from 95.3% (medical) to 99.4% (diagnostical and therapeutical). Those who used institutional sources for information tended to advice vaccination more frequently (95.2%) then those who did not (87.2%). Moreover, 97.4% of those who received a flu vaccination in the previous vaccination campaign tended to recommend the COVID-19 vaccine to relatives, compared to 83.3% of those who did not receive it. Interestingly, not having received a diagnosis of COVID-19 influenced the tendency to recommend vaccination to one’s relatives to a greater extent than not having received it (95.4% vs. 90.9%). Among those advising the COVID-19 vaccine to their relatives, the mean value of the score referring to their fear of being infected by COVID-19 was 7.2 out of 10, while those who did not reported a much lower average score (3.7 out of 10). Finally, those who recommended a vaccination to their relatives and those who did not scored similarly when considering their knowledge about COVID-19 vaccines (7.6 vs. 7.0 out of 10).

When considering exclusively physicians (Table 5), the multivariable logistic regression underlined a weak positive association between their sense of fear and their tendency to advise COVID-19 vaccines both to patients (odds ratio [OR] 1.4; 95% confidence interval [95%CI] 1.2, 1.5) and relatives (OR 1.4; 95%CI 1.2, 1.5); a weak positive association was also described between advising COVID-19 vaccines to one’s relatives and a physicians’ mean level of knowledge on COVID-19 vaccines (OR 1.3; 95%CI 1.1, 1.5). A moderate positive association was found between physicians having received a flu jab in the previous vaccination campaign and their tendency to advise COVID-19 vaccines, respectively, to their patients (OR 5.3; 95%CI 2.9, 9.9) and relatives (OR 4.5; 95%CI 2.5, 8.3). A strong positive association was described between the physicians advising the COVID-19 vaccine to their relatives and being employed in diagnostic and therapeutic departments (OR 8.5; 95%CI 1.1, 67.1).

As for dentists (Table 6), the multivariable logistic regression underlined a weak modest association between a dentists’ fear of COVID-19 and advising COVID-19 vaccines, respectively, to their patients (OR 1.5; 95% CI 1.3, 1.8) and to their relatives (OR 1.6; 95% CI 1.3, 2.0). A strong positive association was described between advising the COVID-19 vaccine to their patients and having received a flu vaccine in the previous vaccination campaign (OR 10.5; 95% CI 2.7, 39.9). Finally, a very strong positive association was highlighted between advising the COVID-19 vaccine to relatives and having received a flu vaccine in the previous vaccination campaign (OR 25.0; 95% CI 4.3, 146.6).

When considering both physicians and dentists (Table 7), the multivariable logistic regression showed a weak positive association between their sense of fear and their tendency to advise COVID-19 vaccines to their patients (OR 1.4; 95%CI 1.3, 1.5), and a moderate positive association between being a dentist and advising COVID-19 vaccines to their patients (OR 2.5; 95%CI 1.3, 4.7); in addition, a moderate to strong positive association could be identified between physicians or dentists and having received a flu jab in the previous vaccination campaign and their tendency to advise COVID-19 vaccines to their patients (OR 6.2; 95%CI 3.6, 10.6). Moreover, a weak positive association was described between physicians and dentists advising the COVID-19 vaccine to their relatives and, respectively, their mean level of knowledge on COVID-19 vaccines (OR 1.2; 95%CI 1.0, 1.4) and their fear of COVID-19 (OR 1.4; 95%CI 1.3, 1.5). Lastly, a moderate and a strong positive association was found between being a dentist and advising the COVID-19 vaccine to their patients (OR 2.5; 95%CI 1.3, 4.7) and to their relatives (OR 2.1; 95%CI 1.1, 4.0). Moreover, a strong positive association was described between having received a flu vaccine in the previous vaccination campaign and advising the COVID-19 vaccine to their patients (OR 6.2; 95%CI 3.6, 10.6) and to their relatives (OR 6.0; 95%CI 3.5, 10.3).

## 4. Discussion

In this study we analyzed the attitudes of the physicians and dentists of the Lazio region towards COVID-19 vaccination, their determinants, and their training needs. In particular, we found a great adherence to COVID-19 vaccination (strongly related to the government mandate), albeit for different reasons. Physicians and dentists are an important source of health and preventive information and immunization advice for their patients [34,35,36]. Indeed, patients show a strong trust in the opinions of their physicians, who can influence their decisions and behaviors [35,37]. Therefore, it is critical to fully understand their attitudes toward COVID-19 vaccination [35,46]. Addressing the reasons and factors associated with possible vaccine hesitancy in these categories of HCWs is the starting point for dismantling false beliefs and misconceptions, which would negatively affect patients’ choices and hinder the fight against the pandemic [47,48]. In contrast, these healthcare professionals can play a crucial role in spreading a confidence in vaccination among patients and overcoming resistance to the vaccine by promoting an adherence to the vaccination schedules [15,37,49,50,51].

Our survey showed an adherence to vaccination by physicians and dentists of 99%. Such a high value was certainly influenced by the institution of mandatory vaccination for HCWs by the Italian Government’s Decree-Law 1 April 2021, n. 44: this mandate resulted in those who refused vaccination being banned from practicing, with a possible relocation to other jobs or an unpaid suspension from work [52]. The literature reports conflicting data regarding physicians and dentists’ acceptance of a COVID-19 vaccination [35,49,53,54,55]. Regarding physicians, Neumann-Böhme et al. showed a median overall acceptance rate of 88.6% when considering 19 countries; it ranged from 62% to 80% in Europe, was 36% in Singapore, and 57.6% in the US [56,57,58]. Our results are similar to those of Sirikalyanpaiboon et al. [35], who also reported a very high vaccination adherence among physicians (95.7%), even though in their case there was no vaccination requirement for HCWs. Bartos et al. [53], in a survey of nearly 10,000 physicians, also showed that 90% of respondents intended to vaccinate and that 89% had confidence in the approved vaccines. According to the American Medical Association, as of January 2022, more than 96% of physicians in the US were vaccinated, and those not yet vaccinated intended to do so [59]. In contrast, other studies have described a greater tendency for vaccine hesitancy (almost 19%) among physicians than among other categories of HCWs [60,61]. Considering dentists instead, one study found that more than 82% of the dentists surveyed were in favor of vaccination [49], similar to what Lin et al. reported [62].

Considering the reasons for vaccination, most of the HCWs included in our survey believed in vaccines as a public health tool to deal with pandemics, more than half were vaccinated out of a sense of responsibility and because they believed they were in a high-risk profession, while a smaller percentage wanted to protect their families; the latter motivation was also found among dentists by Belingheri et al. [49]. In contrast, according to our study, the main reasons for refusing to undergo vaccination were insufficient evidence about its efficacy and safety, a fear of the possible side effects, a belief that they were not at risk of developing severe COVID-19, having medical conditions that contraindicate vaccine administration, and not believing that COVID-19 is a serious disease. These reasons are typically related to vaccine hesitancy and have also been widely described by other studies in the literature [35,47,60,61,63,64,65]. As for the safety concerns and a fear of the adverse effects, both short- and long-term, they could be related to the rapidity of the development of the vaccine [66] and the initial paucity of the safety data available at the beginning of the vaccination campaign [49,67]. In addition, vaccine hesitancy has been found to be higher among physicians who had developed adverse effects at previous vaccinations [68]. In other studies, physicians who believed they had a high risk of contracting COVID-19 and developing serious complications were more likely to accept the vaccine, demonstrating how risk perception influences one’s attitudes toward vaccination [60,69]. In contrast, physicians and dentists who had been diagnosed with COVID-19 in the past were more frequently hesitant: this attitude may stem from the belief that having acquired natural immunity through infection makes vaccination less useful and exposes them to a greater risk of the adverse effects [60,68].

Moreover, in our study, 76% of HCWs did not receive the flu vaccine in the previous campaign, a finding in line with that reported by Sirikalyanpaiboon et al. [35]. Several studies have shown an association between an adherence to the flu vaccination program in previous seasons and attitudes toward COVID-19 vaccination, suggesting that attitudes toward the COVID-19 vaccine mirror the attitudes towards vaccines in general [49,54,62,70]. As for physicians, the results of our survey also highlighted the existence of a positive association between having received the flu vaccine in the previous vaccination campaign and the tendency to recommend the COVID-19 vaccine to their patients and relatives.

In addition, evaluating the responses of both physicians and dentists, it was found that 97.4% of those who had received the flu vaccination in the previous vaccination campaign would recommend the COVID-19 vaccine to their patients, compared with 83.3% of those who had not received it. Overall, 94% of the providers interviewed in our survey recommended the vaccine to their relatives, while 91% recommended it to their patients. Comparing the tendency to recommend the vaccine to patients, the percentage of those who had not recommended it was higher among physicians (12.4%) compared to dentists (4.1%). One worrying aspect we found is that only 89.8% of the GPs interviewed recommended the vaccine to their relatives and only 91.2% recommended it to their patients. In this regard, Gesser-Edelsburg et al. found in their study that the main motivation for family physicians to vaccinate was the perception of institutional pressures rather than desire to protect their patients [71].

In order to provide correct patient information, it is imperative that healthcare providers update their knowledge based on the scientific evidence. For this purpose, institutions play a key role, which must assume unambiguous modes of communication and disseminate the reliable and verified data [72,73,74,75,76]. Therefore, it will be of great interest to further analyze what sources health professionals use to acquire further knowledge about vaccines and how these affect their attitudes. In this context, it is alarming evidence that a high percentage of HCWs do not fully believe in the evidence regarding the COVID-19 vaccines disseminated by pharmaceutical companies and national institutions [43,77,78,79]. Misinformation, particularly about the efficacy and safety, appears to be a major cause of the distrust of vaccines [35,65,80]. In this regard, studies report that physicians who had sought information primarily on social media tended to be more hesitant about vaccines [35,81]. In contrast, the physicians and dentists included in our survey stated that they mainly consulted institutional websites and scientific articles and attended webinars and refresher courses about COVID-19 vaccines; to a lesser extent, they consulted colleagues or consulted newspapers, magazines, television, radio, or other websites/social media. We found that respondents who preferred institutional sources of information tended to recommend vaccination to patients more frequently (95.2%). To encourage positive attitudes toward vaccination, it would be appropriate to educate HCWs on how to recognize and seek reliable, evidence-based information [82]. In addition, government agencies and the health authorities should strive to incorporate the training needs of these HCWs and prepare specific interventions, in order to increase vaccination coverage [68,83,84].

In our study, we found that a large proportion of the surveyed healthcare professionals would like to receive more information about the efficacy, safety, and side effects of the COVID-19 vaccines through institutional websites, refresher courses, and webinars, if possible. Therefore, based on our findings, it might be beneficial for the health authorities to adopt a more transparent communication approach to physicians and dentists and to address the communication barriers and misinformation that might hinder vaccine acceptance [50,55,61,65,73,74]. For this purpose, it might be desirable in the future to further investigate the main concerns and reasons for vaccine hesitancy in physicians and dentists in order to implement specific strategies, such as ad hoc educational interventions, and disseminate the most up-to-date scientific evidence on the vaccine [49,55,65,83].

Finally, our study has some limitations. In particular, the high adherence to the COVID-19 vaccination is partly justified by the government-imposed vaccination requirement. However, in our study, we detail the reasons and attitudes about the vaccination, regardless of the obligation. In addition, our survey exploited a web-based platform, thus reducing the sample to only the users (albeit now the majority) of such tools. Moreover, having disseminated the questionnaire through an online survey, we were not able to calculate the reachable audience and therefore it was not possible to calculate the response rate. Likewise, although our study is limited to only one region, the physicians and dentists working there are more than 7% of the national total, thus providing reliable estimates. Another limitation, considering the survey structure, is that we considered as negative all the responses in which respondents merely recommended the vaccine only to the frailest patients. However, dividing these responses into multiple categories would have resulted in too low a numerosity in an ordinal logistic regression or multinomial logistic regression. Finally, during the period of the administration of the questionnaire (about 5 months), the knowledge and expertise regarding COVID-19 vaccines increased significantly, as did the data regarding their safety and efficacy, thus representing the factors that may have influenced the responses.

## 5. Conclusions

Physicians and dentists are an important source of health information for patients and those from the Lazio region in Italy proved a great adherence to COVID-19 vaccination. They described their main reasons for COVID-19 vaccine refusal, and we suggest that the attitudes toward the COVID-19 vaccine might mirror the attitudes toward vaccines in general. In order to provide patients with correct information, healthcare providers should update their knowledge based on the scientific evidence. For this reason, institutions must adopt unambiguous modes of communication and spread reliable and verified data. Addressing the reasons for the vaccine hesitancy of healthcare professionals could guide the health authorities in adopting the best communication approach and in overcoming the communication barriers.

## Figures and Tables

**Table 1 vaccines-10-01793-t001:** Dichotomic encoding for Question 33 and Question 34.

Q33. Are you recommending COVID-19 vaccine to your patients?	Q34. Are you recommending COVID-19 vaccine to your family members?	Dichotomic encoding
Yes, to everyone	Yes, to everyone	Yes (1)
Yes, but only to the most fragile patients	Yes, but only to the most fragile family members	No (0)
I will wait for more data about efficacy and safety	I will wait for more data about efficacy and safety	No (0)
It depends on the type of vaccine	It depends on the type of vaccine	No (0)
No, I do not recommend it	No, I do not recommend it	No (0)
I do not know	I do not know	No (0)
Not applicable		Missing (.)

**Table 2 vaccines-10-01793-t002:** Descriptive analysis.

Variable		Frequency
Sex	M	810 (53.1%)
	F	711 (46.6%)
	Not specified	5 (0.3%)
Age	24–29	87 (5.6%)
	30–64	901 (59.1%)
	>65	538 (35.3%)
Cohabitation	Not cohabitant	399 (26.2%)
	Cohabitant	1099 (72.0%)
	Not specified	28 (1.,8%)
Level of education	Master’s degree	441 (28.9%)
	Specialization or PhD	1085 (71.1%)
Health profession	Physician	1363 (89.3%)
	Dentist	163 (10.7%)
Work setting	Hospital	340 (22.3%)
	Teaching hospital	188 (12.3%)
	Specialist outpatient clinic	120 (7.9%)
	Private activity	620 (40.6%)
	General practice	163 (10.7%)
	Public health district	129 (8.45%)
	Extended care	534 (2.2%)
	Home care	28 (1.8%)
	Outpatient first aid	12 (0.80%)
	Research institution	42 (2.75%)
	Other	211 (13.8%)
Work department (physicians)	Medicine	605 (39.3%)
	Surgery	194 (12.6%)
	Diagnosis and Therapy	190 (12.4%)
	Public health	173 (11.2%)
	General Practice	149 (9.7%)
	Other	228 (14.8%)
Chronic diseases	Yes	541 (35.4%)
	No	985 (64.6%)
Previous SARS-CoV-2 infection	Yes, asymptomatic	26 (1.7%)
	Yes, mild–moderate symptoms	106 (6.9%)
	Yes, severe symptoms and hospitalization	16 (1.1%)
	No	1378 (90.3%)

**Table 3 vaccines-10-01793-t003:** Descriptive analysis of the distribution of respondents according to their tendency to recommend vaccination to patients.

Variable		Advice to Patients	
		No (%)	Yes (%)
Sex	M	3.7	96.3
	F	6.1	93.9
Age	24–29	2.4	97.6
	30–64	5.4	94.6
	>65	4.6	95.4
Cohabitation	Not cohabitant	6.0	94.0
	Cohabitant	4.6	95.4
Level of education	Master’s degree	5.7	94.4
	Specialization or PhD	4.7	95.3
Health profession	Physician	12.6	87.4
	Dentist	4.1	95.9
Work setting	Hospital	6.1	94.0
	Out-of-hospital	2.9	97.1
Work department (physicians)	Medicine	4.0	96.1
	Surgery	4.0	96.0
	Diagnosis and Therapy	0.6	99.4
	Public health	2.6	97.4
	General Practice	8.8	91.2
	Other	4.7	95.3
Institutional sources for information	No	12.8	87.2
	Yes	4.8	95.2
Social media for information	No	4.9	95.1
	Yes	7.7	92.3
Colleagues for information	No	5.2	94.9
	Yes	4.7	95.3
Flu vaccination (in the previous campaign)	No	16.7	83.3
	Yes	2.6	97.4
Chronic disease(s)	No	5.3	94.7
	Yes	4.4	95.6
Previous diagnosis of COVID-19	No	4.7	95.3
	Yes	7.8	92.3
**Variable**		**Advice to patients**	
		**No**	**Yes**
Fear of being infected by COVID-19	Median	3.0	8.0
	Interquartile range	4.0	5.0
Knowledge about COVID-19 vaccines	Median	7.0	8.0
	Interquartile range	2.0	1.0

**Table 4 vaccines-10-01793-t004:** Descriptive analysis of the distribution of respondents according to their tendency to recommend vaccination to relatives.

Variable		Advice to Relatives	
		No (%)	Yes (%)
Sex	M	4.1	95.9
	F	5.9	94.1
Age	24–29	1.2	98.8
	30–64	5.7	94.3
	>65	4.6	95.4
Cohabitation	Not cohabitant	6.0	94.0
	Cohabitant	4.7	95.3
Level of education	Master’s degree	5.9	94.1
	Specialization or PhD	4.7	95.3
Health profession	Physician	11.3	88.7
	Dentist	4.3	95.7
Work setting	Hospital	6.3	93.7
	Out-of-hospital	2.7	97.3
Work department (physicians)	Medicine	4.7	95.3
	Surgery	4.0	96.0
	Diagnosis and Therapy	0.6	99.4
	Public health	1.8	98.3
	General Practice	10.2	89.8
	Other	3.7	96.3
Sources of information (institutional sources)	No	12.8	87.2
	Yes	4.8	95.2
Sources of information (social media)	No	5.0	95.0
	Yes	5.8	94.2
Sources of information (colleagues)	No	5.5	94.5
	Yes	4.3	95.7
Flu vaccination (in the previous campaign)	No	16.7	83.3
	Yes	2.6	97.4
Chronic disease(s)	No	5.7	94.3
	Yes	3.9	96.2
Previous diagnosis of COVID-19	No	4.6	95.4
	Yes	9.2	90.9
**Variable**		**Advice to relatives**	
		**No**	**No**
Fear of being infected by COVID-19	Median	2.5	8.0
	Interquartile range	4.0	5.0
Knowledge about COVID-19 vaccines	Median	7.0	8.0
	Interquartile range	2.0	1.0

**Table 5 vaccines-10-01793-t005:** Multivariable logistic analysis: physicians.

Socio-Demographic Variables		Advising COVID-19 Vaccine to Patients (95% CI)	Advising COVID-19 Vaccine to Relatives (95% CI)
Sex	Male	1.0 (1.0, 1.0)	1.0 (1.0, 1.0)
	Female	0.5 (0.3, 0.1)	0.7 (0.4, 1.4)
Age		1.0 (1.0, 1.0)	1.0 (1.0, 1.0)
Specialization	Clinical specialties	1.0 (1.0, 1.0)	1.0 (1.0, 1.0)
	Surgical specialties	1.0 (0.4, 2.5)	1.2 (0.5, 3.0)
	Diagnostic/therapeutic services specialties	6.8 (0.9, 53.9)	8.5 * (1.1, 67.1)
	Public health services specialties	1.2 (0.3, 4.2)	2.4 (0.5, 10.9)
	General practice	0.5 (0.2, 1.1)	0.5 (0.2, 1.1)
	Other	0.8 (0.3, 1.8)	1.4 (0.5, 3.6)
Working setting	Hospital	1.0 (1.0, 1.0)	1.0 (1.0, 1.0)
	Other than hospital	1.3 (0.6, 2.8)	1.7 (0.8, 3.5)
**COVID-19-related variables**			
Fear of COVID-19		1.4 *** (1.2, 1.5)	1.4 *** (1.2, 1.5)
Knowledge about COVID-19 vaccines		1.2 (1.0, 1.4)	1.3 ** (1.1, 1.5)
**Information sources about COVID-19 vaccines**			
Institutional sources	No	1.0 (1.0, 1.0)	1.0 (1.0, 1.0)
	Yes	1.9 (0.5, 7.4)	1.9 (0.5, 7.7)
**Health-related variables**			
Previous flu vaccination	No	1.0 (1.0, 1.0)	1.0 (1.0, 1.0)
	Yes	5.3 *** (2.9, 9.9)	4.5 *** (2.5, 8.3)
Previous SARS-CoV-2 infection	No	1.0 (1.0, 1.0)	1.0 (1.0, 1.0)
	Yes	0.9 (0.4, 2.0)	0.7 (0.3,1.4)
	N	1305	1305

Exponentiated coefficients; 95% confidence intervals in brackets. * *p* < 0.05, ** *p* < 0.01, *** *p* < 0.001.

**Table 6 vaccines-10-01793-t006:** Multivariable logistic analysis: dentists.

*Socio-Demographic Variables*		Advising COVID-19 Vaccine to Patients (95% CI)	Advising COVID-19 Vaccine to Relatives (95% CI)
Sex	Male	1.0 (1.0, 1.0)	1.0 (1.0, 1.0)
	Female	0.6 (0.1, 2.8)	0.6 (0.1, 3.8)
Age		1.0 (0.9, 1.1)	1.0 (0.9, 1.0)
**COVID-19-related variables**			
Fear of COVID-19		1.5 *** (1.3, 1.8)	1.6 *** (1.3, 2.0)
Knowledge about COVID-19 vaccines		1.0 (0.6, 1.6)	0.8 (0.5, 1.4)
**Information sources about COVID-19 vaccines**			
Institutional sources	No	1.0 (1.0, 1.0)	1.0 (1.0, 1.0)
	Yes	0.9 (0.04, 22.4)	1.4 (0.06, 33.7)
**Health-related variables**			
Previous flu vaccination	No	1.0 (1.0, 1.0)	1.0 (1.0, 1.0)
	Yes	10.5 *** (2.7, 39.9)	25.0 *** (4.3, 146.6)
Previous SARS-CoV-2 infection	No	1.0 (1.0, 1.0)	1.0 (1.0, 1.0)
	Yes	3.0 (0.3, 28.9)	2.4 (0.2, 25.7)
	N	159	159

Exponentiated coefficients; 95% confidence intervals in brackets. *** *p* < 0.001.

**Table 7 vaccines-10-01793-t007:** Multivariable logistic analysis: physicians and dentists.

*Socio-Demographic Variables*		Advising COVID-19 Vaccine to Patients (95% CI)	Advising COVID-19 Vaccine to Relatives (95% CI)
Sex	Male	1.0 (1.0, 1.0)	1.0 (1.0, 1.0)
	Female	0.6 (0.3, 1.0)	0.7 (0.4, 1.2)
Age		1.0 (1.0, 1.0)	1.0 (1.0, 1.0)
Professional category	Physician	1.0 (1.0, 1.0)	1.0 (1.0, 1.0)
	Dentist	2.5 ** (1.3, 4.7)	2.1 * (1.1, 4.0)
** *COVID-19-related variables* **			
Fear of COVID-19		1.4 *** (1.3, 1.5)	1.4 *** (1.3, 1.5)
Knowledge about COVID-19 vaccines		1.1 (1.0, 1.3)	1.2 * (1.0, 1.4)
** *Information sources about COVID-19 vaccines* **			
Institutional sources	No	1.0 (1.0, 1.0)	1.0 (1.0, 1.0)
	Yes	1.5 (0.4, 5.3)	1.4 (0.4, 5.0)
** *Health-related variables* **			
Previous flu vaccination	No	1.0 (1.0, 1.0)	1.0 (1.0, 1.0)
	Yes	6.2 *** (3.6, 10.6)	6.0 *** (3.5, 10.3)
Previous SARS-CoV-2 infection	No	1.0 (1.0, 1.0)	1.0 (1.0, 1.0)
	Yes	1.2 (0.5, 2.5)	0.9 (0.4, 1.9)
	N	1464	1464

Exponentiated coefficients; 95% confidence intervals in brackets. * *p* < 0.05, ** *p* < 0.01, *** *p* < 0.001.

## Data Availability

The data presented in this study are available upon reasonable request from the corresponding author.

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
