# Peer review of "Factors Influencing the Choice to Advise for or against COVID-19 Vaccination in the Physicians and Dentists of an Italian Region"

_vaccines, 2022, doi:10.3390/vaccines10111793_

Round 1

Reviewer 1 Report

Dear authors,

Your article i quite interesting since the hesitancy against vaccination stands as a crucial theme in relevance to public health and especially when this hesitance finds positive ground among health care professionals.

The abstract is well structured and points out the main findings. The introduction is concrete, however I would recommend to add a paragraph in relevance to differences regarding vaccination hesitance as well as knowledge differences among different categories of health care professionals (physicians, doctors, physiotherapists, dentists).

The material and methods are adequately presented. The sample size is satisfactory and the selection method is acceptable (online survey). Informed consent was granted and ethical approval was provided by the Ethics Committee. The response rate was high (71,6%).

The description of the questionnaire is satisfactory as well as the statistical analysis section.

The results are fairly presented and the tables are very helpful. In the discussion it is commended that the participants showed great level of adherence towards COVID-19 Vaccination.

The discussion is focused on the main findings and is well referenced. The determinants that influences participants' decision regarding vaccination are quite addressed.

The limitations of the study are discussed. But it is not clear whether the 28,4% of the health workers that did not participate the study would share the same beliefs and attitudes. I would recommend to add this limitation since the generalizability of the results even among the health professionals in the specific setting are questioned in a way.

The conclusions are in line with the findings.

Author Response

  1. The abstract is well structured and points out the main findings.

  1. The introduction is concrete, however I would recommend to add a paragraph in relevance to differences regarding vaccination hesitance as well as knowledge differences among different categories of health care professionals (physicians, doctors, physiotherapists, dentists).

We thank the reviewer for the remark. We modified the “Introduction” paragraph accordingly (see lines 67-72).

  1. The material and methods are adequately presented. The sample size is satisfactory and the selection method is acceptable (online survey). Informed consent was granted and ethical approval was provided by the Ethics Committee. The response rate was high (71,6%).

We thank the reviewer for the remark. The sample of physicians and dentists analyzed in the present study (n.1526) is part of a sample of healthcare workers who completed the online survey in the original study (n.2132). Since ours is an online questionnaire, it is not possible for us to calculate the number of people who received the questionnaire and to know what the response rate was.

  1. The description of the questionnaire is satisfactory as well as the statistical analysis section.

The results are fairly presented and the tables are very helpful.

In the discussion it is commended that the participants showed great level of adherence towards COVID-19 Vaccination.

The discussion is focused on the main findings and is well referenced. The determinants that influences participants' decision regarding vaccination are quite addressed.

  1. The limitations of the study are discussed. But it is not clear whether the 28,4% of the health workers that did not participate the study would share the same beliefs and attitudes. I would recommend to add this limitation since the generalizability of the results even among the health professionals in the specific setting are questioned in a way.

We thank the reviewer for this suggestion. Regarding the question of response rate, we have responded to point 3. However, we take up this suggestion and we include as a limitation the fact that since we have disseminated the questionnaire through an online survey, we do not know what the reachable audience is, and therefore we cannot calculate the real response rate. In addition, as already reported in the limitations, this way of dissemination limited the survey completion to only those who had access and ability to use digital tools (see lines 415-419).

Reviewer 2 Report

Please write a paragraph about the representativnes of your sample by comparing demographic characteristics of your sample with the characteristiscs of all doctors in the region you analyze

Author Response

Please write a paragraph about the representativeness of your sample by comparing demographic characteristics of your sample with the characteristiscs of all doctors in the region you analyze

We thank the reviewer for the suggestion. Unfortunately, data on the sociodemographic characteristics of physicians and dentists residing in Lazio have not been made public either by the Italian Statistical Institute (Istat) or the professional orders. Therefore, it is not possible for us to make a comparison between our sample and all physicians and dentists in the Lazio region.

Reviewer 3 Report

Dear Authors

In this study authors analyzed the attitudes of physicians and dentists of Lazio Region 

toward COVID-19 vaccination, their determinants, and their training needs. Authors found a great adherence to COVID-19 vaccination (strongly related to the government mandate) albeit for different reasons. The survey showed an adherence to vaccination by physicians and dentists of 99%.

It is recognized that physicians and dentists are an important source of health and preventive information  and immunization advice for patients, since they show strong trust in the opinions of physicians, who can influence their decisions and behaviors. So, it is critical to fully understand their attitudes toward COVID-19 vaccination.

The paper is well written and interesting to read. The overall manuscript presentation was impressive and interesting. Title is clear and informative; it displays the main objective of the study. The abstract is sectioned. It contains focused background with clear objective. The literature review follows the specific aim of the study. The tables have sufficient, good quality and appropriately illustrative of the paper contents. The study from a scientific point of view seems to be well done and presents good results, from where to derive valid conclusions. Introduction summarizes relevant research to provide context and clearly state the problem.  The research methods used ensure the reliability of the obtained results. The discussion section interprets the findings in view of the results obtained in this and in past studies on this topic.

References cited are recent and have a high relevance to the problem.

Minor corrections

In material and methods, logistic regression performed is not explain in detail. What type of logistic regression is performed? Materials and methods should be complete enough to allow possible replication of the research. Please, add detailed information.

Author Response

In material and methods, logistic regression performed is not explain in detail. What type of logistic regression is performed? Materials and methods should be complete enough to allow possible replication of the research. Please, add detailed information.

We thank the reviewer for his precious suggestion. We have edited the section as follows: “The logistic regression was applied by fitting a logit model for a binary response by maximum likelihood. The independent variables to be included in the multivariable model have been selected on the basis of preliminary significance (p<0.10) in logistic univariable models. Age has been included in the model despite non-significance to account for potential confounding.” (see lines 151-155).

Reviewer 4 Report

In this manuscript the authors perform a logistic regression in order to determine which factors were associated with the potential recommendation of COVID-19 vaccination in patients and relatives of certain Italian physicians and dentists. I find this article of great relevance as in many cases these healthcare workers are the source of information for many patients, specially in topics such as vaccination. The manuscript is well written and it is easy to understand; however, there are certain aspects that would need to be solved before its acceptance. Congratulations for this nice work.

GENERAL ASPECTS

- I suggest to use the terms "univariaBLE" and "multivariBLE".

- I wonder whether the term "Lazio" might not be better known than "Latium" [just a comment]. And I would suggest to stick to one of the terms and not both, not to confuse the reader. Additionally, I would suggest to mention that it is the region of Rome.

- If you consider "gender" as a sociodemographic term, do not make any changes, if as a biological term, I would suggest to use "sex".

- Before using any abbreviation, please state their definition (in their first use), including "SARS-CoV-2", "COVID-19", etc.

- "region" does not need to be capitalized, neither in the title nor the text.

- Decimals in English are with ".", not ",".

- Could it be possible to provide as an appendix the survey?

INTRODUCTION

- I would suggest to groups it in 3 paragraphs, as in some cases you present 1 sentence = 1 paragraph.

- How much is, if known, "HCWs is a topic already investigated; indeed, insufficient vaccination coverage in Italian health personnel is reported", is it possible to provide numbers?

MATERIALS AND METHODS

- I would suggest to make paragraphs of more than 1 sentence each.

- "Only Lazio Region was considered because our analysis involved Rome Provincial Order of Physicians and Dentists “Omceo”" Does OMCEO rule in the other provinces of Lazio beyond Rome?

- Could it be possible to provide the name and site of the software company/institution in charge of "Survey Monkey®" platform?

- "the link was sent to physicians and dentists by the professional association (Rome Provincial Order of Physicians and Dentists), and via newsletter.". Then, the target participants are only Roman physicians/dentists or Lazians in general?

- "Fondazione Policlinico Universitario A. Gemelli - IRCCS", which is based in?

- Questionnaire structure: you asked for gender or for sex?

- If your sample is non-normally distributed, please use median and interquartile range.

RESULTS

- I wonder how could you recruit, at least, a 18 years-old phyisician or dentist in Italy, considering that, in general, student start at university at the age of 19. Was there any person with high capacity that by age of 18 had already their degree? Additionally, there was a 95 year-old physician/dentist?

- I addition to the above, I would suggest to reclassify the age, starting the range with the youngest person, which definitely will not be 0 years old.

- Why does the percentage in "work setting" sum up to 123%?

- Why in table 3 "marital status" has different categories than in table 2? Should not be "Cohabitance" the name of the variable in table 3?

- No univariable analysis was performed before the multivariable? I would suggest to observe which variables are relevant in the univariable (p<0.1) and then use those in the multivariable analysis.

- Not sure whether it depends on the authors or the editorial, but I would suggest to remove so many lines, it makes pretty complicated to understand the table. Additionally, it could be helpful for the reader if the OR and the 95% CI appear together in the same row. I guess there is space for that considering the blank areas currently available.

- Why is there no sub-analysis for dentists too?

DISCUSSION

- "Government Decree-Law April 1, 2021" of which government? Italian Republic, Lazio region, Rome province?

- I would suggest to not leave 1 sentence=1 paragraph.

- In your results you were providing 2 decimals, but here only one, I would suggest the latter consistently throughout the whole text.

- "Moreover, in our study, 76% of HCWs didn’t receive the flu vaccine". Not a problem for this reviewer, but I guess it should be "did not receive".

- "Likewise, in fact, this platform allowed for a nationwide distribution of the sample, thus considering possible differences related to residence." As far as I see, there is no result on the patient residence. Additionally, if you are providing data only from Lazio, where is the limitation? Please, provide an extended explanation to this. Which HCW have participated in your survey? Only those working in Lazio, living in Lazio, either...

- I would consider a limitation that the response "Yes, [I recommend the vaccine] but only to the most fragile patients" is considered fully as a no, when actually can be either yes or no. I would say this was a limitation originated in the moment of the survey conception.

Author Response

GENERAL ASPECTS

  1. I suggest to use the terms "univariaBLE" and "multivariBLE".

I wonder whether the term "Lazio" might not be better known than "Latium" [just a comment]. And I would suggest to stick to one of the terms and not both, not to confuse the reader. Additionally, I would suggest to mention that it is the region of Rome.

If you consider "gender" as a sociodemographic term, do not make any changes, if as a biological term, I would suggest to use "sex".

Before using any abbreviation, please state their definition (in their first use), including "SARS-CoV-2", "COVID-19", etc.

"region" does not need to be capitalized, neither in the title nor the text.

Decimals in English are with ".", not ",".

We thank the reviewer for these remarks. We amended the text accordingly.

  1. Could it be possible to provide as an appendix the survey?

We thank the reviewer for the suggestion. We attached the questionnaire as an appendix.

INTRODUCTION

  1. I would suggest to groups it in 3 paragraphs, as in some cases you present 1 sentence = 1 paragraph.

We thank for the suggestion. We modified the “Introduction” paragraph accordingly (see lines 33-76).

  1. How much is, if known, "HCWs is a topic already investigated; indeed, insufficient vaccination coverage in Italian health personnel is reported", is it possible to provide numbers?

    We thank the reviewer for the remark. We modified the “Introduction” paragraph accordingly (see lines 64-67).

MATERIALS AND METHODS

  1. I would suggest to make paragraphs of more than 1 sentence each.

We thank for the suggestion. We modified the “Methods” paragraph accordingly (see lines 77-165).

  1. "Only Lazio Region was considered because our analysis involved Rome Provincial Order of Physicians and Dentists “Omceo”" Does OMCEO rule in the other provinces of Lazio beyond Rome?

We thank the reviewer for this remark. We chose to conduct the analysis at the regional level because it was the smallest geographic level obtainable based on the available data. Omceo Rome was the main form of dissemination of the questionnaire, however, the questionnaire also reached members of professional orders in the other provinces of Lazio region through word of mouth and through flyers spread during the national congress of the Italian Society of Hygiene, Preventive Medicine and Public Health (see lines 92-99).

  1. Could it be possible to provide the name and site of the software company/institution in charge of "Survey Monkey®" platform?

We thank the reviewer for the remark. We added the name of the software company in the text (see line 93).

  1. "the link was sent to physicians and dentists by the professional association (Rome Provincial Order of Physicians and Dentists), and via newsletter.". Then, the target participants are only Roman physicians/dentists or Lazians in general?

We thank the reviewer for the observation. As also specified in point n.11, the target participants were all physicians and dentists residing in Lazio region.

  1. "Fondazione Policlinico Universitario A. Gemelli - IRCCS", which is based in?

We thank the reviewer for the note. We specified the location in the text (see line 105).

  1. Questionnaire structure: you asked for gender or for sex?

We thank the reviewer for the observation. We asked for sex in the questionnaire, so we have standardized the word throughout the manuscript.

  1. If your sample is non-normally distributed, please use median and interquartile range.

We thank the reviewer for the remark. We have edited the tables and text by using the median and interquartile range.

RESULTS

  1. I wonder how could you recruit, at least, a 18 years-old phyisician or dentist in Italy, considering that, in general, student start at university at the age of 19. Was there any person with high capacity that by age of 18 had already their degree? Additionally, there was a 95 year-old physician/dentist?

I addition to the above, I would suggest to reclassify the age, starting the range with the youngest person, which definitely will not be 0 years old.

We thank the reviewer for this precious remark. We modified the age classes so that the extremes correspond to the ages of the youngest and oldest respondents to our survey. The youngest age recorded is 24 years and is consistent with completing a degree in dentistry in Italy.

  1. Why does the percentage in "work setting" sum up to 123%?

We thank the reviewer for the remark. As in the questionnaire more than one answer was possible, the total exceeds 100%.

  1. Why in table 3 "marital status" has different categories than in table 2? Should not be "Cohabitance" the name of the variable in table 3?

We thank the reviewer for the comment. We standardized the variable "cohabitance" in all tables.

  1. No univariable analysis was performed before the multivariable? I would suggest to observe which variables are relevant in the univariable (p<0.1) and then use those in the multivariable analysis.

We thank the reviewer for this precious remark. We have included in the model all the variables that we expected might show an association based on literature and on our previous experience. However, after careful consideration, we have decided to accept the suggestion of the reviewer, as we judged it more robust from a statistical standpoint. We have consequently simplified the models and edited the relevant paper sections accordingly (see lines 249-298).

  1. Not sure whether it depends on the authors or the editorial, but I would suggest to remove so many lines, it makes pretty complicated to understand the table. Additionally, it could be helpful for the reader if the OR and the 95% CI appear together in the same row. I guess there is space for that considering the blank areas currently available.

We thank the reviewer for this hint. We modified the tables accordingly.

  1. Why is there no sub-analysis for dentists too?

We thank the reviewer for this observation. Originally, the sub-analysis for physicians had been conceived to be able to consider also the physician-specific variables “work setting” and “work department”. However, we acknowledge that a sub-analysis for dentists could still be interesting, so we have added it (see lines 270-278).

DISCUSSION

  1. "Government Decree-Law April 1, 2021" of which government? Italian Republic, Lazio region, Rome province?

We thank the reviewer for the observation. We edited the sentence accordingly (see line 317).

  1. I would suggest to not leave 1 sentence=1 paragraph.

We thank the reviewer for the suggestion. We modified the “Discussion” paragraph accordingly (see lines 300-428).

  1. In your results you were providing 2 decimals, but here only one, I would suggest the latter consistently throughout the whole text.

We thank the reviewer for the remark. We modified the whole text accordingly.

  1. "Moreover, in our study, 76% of HCWs didn’t receive the flu vaccine". Not a problem for this reviewer, but I guess it should be "did not receive".

We thank the reviewer for the remark. We modified the sentence accordingly (see line 357).

  1. "Likewise, in fact, this platform allowed for a nationwide distribution of the sample, thus considering possible differences related to residence." As far as I see, there is no result on the patient residence. Additionally, if you are providing data only from Lazio, where is the limitation? Please, provide an extended explanation to this. Which HCW have participated in your survey? Only those working in Lazio, living in Lazio, either...

We thank the reviewer for this observation. We actually explained this passage in a confusing way. Our sample consists of physicians and dentists exclusively residing in Lazio region during the pandemic. We specified this point in the limitations: although our study is limited to only one region, the physicians and dentists working there are more than 7% of the national total, thus providing reliable estimates (see lines 418-420). (here you can find the data about the total number of physicians and dentists working in Italy and in the Lazio region: http://dati.istat.it/Index.aspx?QueryId=31546 and https://www.istat.it/it/files//2020/05/12_Lazio_Scheda_rev.pdf).

  1. I would consider a limitation that the response "Yes, [I recommend the vaccine] but only to the most fragile patients" is considered fully as a no, when actually can be either yes or no. I would say this was a limitation originated in the moment of the survey conception.

We thank the reviewer for the observation. We have included it as a limitation as following: “Another limitation, considering the survey structure, is that we considered as negative all the responses in which respondents merely recommended the vaccine only to the frailest patients. However, dividing these responses into multiple categories would have resulted in too low a numerosity to consider it a separate category in an ordinal logistic regression or Multinomial logistic regression.” (see lines 421-425).
